# Inhaled Corticosteroids and the Lung Microbiome in COPD

**DOI:** 10.3390/biomedicines9101312

**Published:** 2021-09-24

**Authors:** Holly R. Keir, Marco Contoli, James D. Chalmers

**Affiliations:** 1Division of Molecular and Clinical Medicine, School of Medicine, University of Dundee, Dundee DD1 9SY, UK; j.chalmers@dundee.ac.uk; 2Department of Translational Medicine, University of Ferrara, 44121 Ferrara, Italy; ctm@unife.it

**Keywords:** COPD, inhaled corticosteroids, lung microbiome

## Abstract

The Global Initiative for Chronic Obstructive Lung Disease 2021 Report recommends inhaled corticosteroid (ICS)-containing regimens as part of pharmacological treatment in patients with chronic obstructive lung disease (COPD) and frequent exacerbations, particularly with eosinophilic inflammation. However, real-world studies reveal overprescription of ICS in COPD, irrespective of disease presentation and inflammatory endotype, leading to increased risk of side effects, mainly respiratory infections. The optimal use of ICS in COPD therefore remains an area of intensive research, and additional biomarkers of benefit and risk are needed. Although the interplay between inflammation and infection in COPD is widely acknowledged, the role of the microbiome in shaping lower airway inflammation has only recently been explored. Next-generation sequencing has revealed that COPD disease progression and exacerbation frequency are associated with changes in the composition of the lung microbiome, and that the immunosuppressive effects of ICS can contribute to potentially deleterious airway microbiota changes by increasing bacterial load and the abundance of potentially pathogenic taxa such as *Streptococcus* and *Haemophilus*. Here, we explore the relationship between microbiome, inflammation, ICS use and disease phenotype. This relationship may inform the benefit:risk assessment of ICS use in patients with COPD and lead to more personalised pharmacological management.

## 1. Introduction

Chronic obstructive pulmonary disease (COPD) is a condition with a heterogeneous phenotype characterised by chronic bronchitis, emphysema and small airways disease, resulting in progressive airflow limitation that is not fully reversible [1,2,3,4]. Globally, COPD presents a growing social and economic burden in terms of both disease prevalence and mortality [5].

According to the Global Initiative for Chronic Obstructive Lung Disease (GOLD) 2021 Strategy Report [6], the goals of treatment in COPD include improvement in exercise capacity and quality of life, reduced exacerbations (which are major drivers of disease progression) and prevention of premature mortality. Non-pharmacological management, including smoking cessation and exercise, is critical. Pharmacological treatment of COPD mainly consists of bronchodilators, i.e., long-acting β_2_-agonists (LABAs) and/or long-acting muscarinic antagonists (LAMAs) [6]. Bronchodilators can improve quality of life, improve lung function, and reduce exacerbation rate in patients with COPD [6]. Inhaled corticosteroid (ICS)-containing regimens are recommended primarily as step-up pharmacological therapy for those patients with COPD who are still experiencing frequent exacerbations despite regular treatments with bronchodilator(s) and who have evidence of eosinophilic inflammation [6].

Despite the GOLD recommendations, evidence from real-world studies suggests that ICS is being over-prescribed in COPD, irrespective of disease presentation and underlying inflammation [7,8,9]. This has important clinical implications, as ICS use is associated with increased risk of side effects, including infection events [10,11,12,13] and particularly with an increased risk of pneumonia [14].

Neutrophil inflammation is the dominant inflammatory endotype of COPD. Airway neutrophilic inflammation is associated with increasing exacerbation frequency, disease severity and mortality in COPD [15,16,17]. Conversely, eosinophilic inflammation accounts for only about 20% of patients when defined by sputum eosinophil counts >3% or blood eosinophils >300 cells/µL [18]. This distinction is important for considering the role of anti-inflammatory therapy in COPD, as corticosteroids can aid the resolution of eosinophilic inflammation through inducing apoptosis of eosinophils. In contrast, neutrophilic disease does not respond to ICS, and evidence suggests that ICS may worsen neutrophilic inflammation by delaying neutrophil apoptosis [19,20], leading to increased secondary necrosis which may release damaging proteases such as neutrophil elastase. This has the potential to impair host defence against infection. Indeed, ICS use has been associated with an increased risk of pneumonia [14], and in vitro and in vivo studies have shown that ICS can impair pulmonary host defences against pathogens [21,22,23,24]. In addition to effects on neutrophils and eosinophils, ICS have multiple anti-inflammatory effects on airway epithelial cells, including downregulation of pro-inflammatory cytokines such as interleukin (IL) 1 beta, granulocyte-macrophage colony-stimulating factor and IL-8, and decreased production of enzymes such as inducible nitric oxide synthase and cyclooxygenase-2, and adhesion molecules including intracellular adhesion molecule 1 [25]. Belvisi and colleagues [26] showed enhanced release of matrix metalloproteinases (MMPs) 1 and 9 and other inflammatory mediators from alveolar macrophages in patients with COPD compared with healthy controls, and Belchamber and coworkers showed that macrophage phagocytosis and clearance of bacteria are impaired in COPD [27]. The release of pro-inflammatory cytokines and MMPs from alveolar macrophages is usually inhibited by corticosteroids but COPD macrophages have been shown to be resistant to corticosteroid inhibition [28,29]. Likewise, impairment in phagocytosis does not appear to be correlated with the use of ICS [30,31].

Airway infections play a central role in the manifestation of the disease, with changes in the composition of the airway microbial community, known as the microbiome, contributing to disease progression and exacerbation frequency [32]. Bacterial and viral infections have been most commonly implicated in the cause of exacerbations, but the extent to which this affects lower airway inflammation and clinical presentation is not completely understood [33,34]. It is widely accepted that the roles of inflammation and infection in COPD are closely linked, and the introduction of next-generation sequencing (NGS) techniques such as 16S rRNA sequencing has enhanced our understanding of the role the lung microbiome plays in disease progression [35].

ICS are immunosuppressive and can affect the interplay between host and microbe, leading to changes in the airway microbiome. For example, in asthma and rhinosinusitis, ICS use has been associated with alteration in the microbiome composition of both the lung and nasal cavity [36,37]. It is also well known that ICS use affects the oral microbiome, predisposing people to fungal infections such as candidiasis [38]. Therefore, identifying and understanding the underlying biological mechanisms behind phenotypes of COPD, during both stable and exacerbating periods, may lead to more targeted and appropriate treatment. This “personalised medicine” approach aims to target ICS treatment to only those patients for whom a benefit:risk ratio is appropriate, rather than a “one size fits all” in which risk may outweigh benefit.

Emerging data on the effects of ICS on the lung microbiome in COPD may help our understanding of the role of ICS, and in this review, we examine recent data suggesting that the lung microbiome may help to inform the benefit:risk assessment associated with the prescription of ICS as well as helping to understand how different patterns of lung inflammation carry different benefits and risks associated with ICS.

## 2. What Do We Know about the Microbiome in COPD?

Charlson et al. in 2011 were the first to conclusively show that the lung in healthy subjects (no history of pulmonary disease or ongoing serious medical illnesses, normal spirometry, and no upper respiratory tract symptoms within 4 weeks) is host to a bacterial microbiome, predominantly the Proteobacteria *Neisseriaceae*, Firmicutes *Streptococcaceae* and *Veillonellaceae*, and Fusobacteria [39]. Dickson and colleagues went on to show that the prevalence and diversity of genera do not significantly differ across lung locations (lingula, right middle lobe, right upper lobe and left upper lobe) [40]. Although present in lower abundance compared with bacteria, a fungal microbiome (the mycobiome) also exists in the healthy airway, with a predominance of *Candida* and *Saccharomyces* [41].

Through classical culture-based methods, it has been shown that the COPD lung can also be host to potentially pathogenic bacteria (e.g., the Proteobacteria *Haemophilus influenzae*, *Moraxella catarrhalis* and *Pseudomonas aeruginosa*, and the Firmicutes *Streptococcus pneumoniae*) [42,43]. It is also firmly established that bacteria play an important role in COPD pathogenesis, with bacterial colonisation correlating with inflammatory response, local immune response and symptoms [44]. Further, pathogenic bacteria contribute to lung damage and loss of lung function, and these pathogens are believed to be responsible for around half of COPD exacerbations [34]. Now, with the advent of NGS technologies, researchers in COPD are able to move beyond culture to further examine how changes in microbiome composition may play a role.

NGS allows massive parallel high-throughput analysis of microbes—the vast majority of which are not cultured routinely in clinical practice [32] (Figure 1). The analysis begins with sample collection (e.g., sputum, bronchoalveolar lavage, oral wash or upper airway swabs), followed by DNA extraction. In the most widely used techniques, polymerase chain reaction (PCR) is then carried out to amplify bacterial or fungal genes. The most commonly used target genes are the 16S rRNA gene, in the case of bacteria [45], or the internal transcribed spacer-1 (ITS1) DNA between the 5.8S, 18S and 28S rRNA genes, in the case of fungi [46], although other approaches are available. These genes consist of both highly conserved and variable regions. The nucleotide sequences of the variable regions of the 16S rRNA gene and ITS1 vary at the genus level, and so sequencing across these regions can be used to provide taxonomic information about members of the bacterial and fungal microbiomes. The resulting amplicon sequences are interrogated against a database of known taxonomic sequences. Sequences are then grouped into operational taxonomic units (OTUs), which can identify members of the microbiome from family to genus level [45]. A more recent technology involves the analysis of sequences for amplicon sequence variants, which offers potentially improved sensitivity and specificity over OTUs [47]. Outputs from these analyses are used to examine the diversity within the sample (the α diversity) and between samples (the β diversity) [48]. The diversity of the microbiome is reflected by the richness and evenness of the populations present within the airway. In disease, there is often a loss of diversity, reflected by the dominance of one or a few genera. In inflammatory lung diseases such as bronchiectasis and cystic fibrosis, diversity has been shown to decrease, and studies indicate that changes in the composition of the lung microbiome might contribute to, and result from, chronic inflammation [49,50,51]. Further, studies have shown relationships between asthma severity and low diversity; for example, severe asthma has been associated with a clear loss of diversity [52], and lower diversity has been associated with neutrophilic asthma, which is often more severe than eosinophilic [53]. In addition, interactions between members of the microbiome can be analysed (whether members co-occur or co-exclude, and the strength of these relationships).

The high-throughput analysis of microbes as described above is relatively cost-effective and can generate data from samples containing organisms that are otherwise rarely cultured. However, the trade-off is that the amplicons generated through widely used 16S rRNA sequencing approaches are too short to provide resolution at the species level, although full-length sequencing of the 16S rRNA gene is possible and is being increasingly used [54,55,56,57]. So-called “shotgun” metagenomic technologies, although not yet commonly used in microbiome analysis in COPD, go further to allow comprehensive sequencing of microbes across the various kingdoms, with the potential to simultaneously sequence bacteria, fungi, viruses and other microbes within a sample [58]. This technique will become more and more commonplace and certainly will be applied to the analysis of the COPD lung microbiome.

Sequencing techniques are not quantitative and therefore analysis of microbiome composition can be supplemented with techniques such as quantitative PCR (qPCR) and droplet digital PCR to determine microbial burden. Primers specific for conserved regions such as those for the 16S rRNA and ITS genes can be used to determine the absolute microbial burden and specific primers used to quantify individual species.

Armed with these powerful tools, the role of the microbiome in COPD and other respiratory diseases is being investigated.

### 2.1. The Microbiome of the COPD Lung versus the Healthy Lung

As indicated above, it is firmly established that the COPD lung is often host to potentially pathogenic bacteria, including *H. influenzae*, *M. catarrhalis, P. aeruginosa* and *S. pneumoniae* [42,43]. Studies have now demonstrated that the composition of the lung microbiome is altered in patients with COPD compared with controls. Einarsson and colleagues [59] compared the lower airway microbiomes of patients with clinically stable COPD and mild-to-severe airflow obstruction, healthy smokers and healthy non-smokers. They found that bronchial wash samples from patients with COPD had a greater prevalence of Proteobacteria (*Haemophilus*) compared with smokers and non-smokers. Haldar and coworkers [60] then went on to compare sputum samples from patients with COPD from the COPDMAP consortium with healthy (no evidence of asthma, COPD or bronchiectasis) smokers and healthy non-smokers. In agreement with Einarsson, it was demonstrated that Proteobacteria (*Haemophilus* and *Moraxella*) was the most frequently dominant taxon in patients with COPD versus healthy controls, in which Firmicutes, Bacteroidetes and Actinobacteria (*Streptococcus*, *Veillonella*, *Prevotella*, *Actinomyces* and *Rothia*) were the dominant taxa. Recently, in a closer examination of the interaction between the lung microbiome and host, Wang and colleagues [61] carried out a large multi-omic meta-analysis of public COPD sputum microbiome datasets, totalling over 1600 samples from across the world. Adjusting for factors such as age, gender and smoking history, *Haemophilus* and *Moraxella* were enriched in stable COPD compared with non-COPD controls, whereas the abundance of genera such as *Campylobacter* and *Prevotella* were reduced. This multi-omic approach was able to reveal that pathways involved in the biosynthesis of bacterial palmitate, homocysteine and urate were upregulated in the COPD microbiome. These metabolites are postulated to have disease-promoting effects: palmitate is known to be associated with enhanced inflammation [62] and oxidative stress [63], and has been observed to be increased in COPD airways previously [64,65].

Taken together, these studies indicate that there is a “COPD lung microbiome” as compared with the non-COPD lung and lower airway, characterised by dominance by Proteobacteria (chiefly *Haemophilus* and *Moraxella*).

### 2.2. The Microbiome in Stable COPD

Although there appears to be a COPD lung microbiome, it is complex and highly distinct between individual patients. In stable disease, studies have demonstrated variations in the microbiome over time within individuals. As such, many questions remain to be answered concerning the precise role of the microbiome in COPD.

In a longitudinal cohort study of clinically stable patients with COPD, Dicker et al. [35] showed that lower microbiome diversity coupled with Proteobacteria dominance (predominantly *Haemophilus*) was associated with more severe COPD, blood eosinophil levels of ≤100/µL and increased mortality rate. Increased blood eosinophil counts were positively associated with the percentage of Firmicutes and *Streptococcus*, and negatively associated with Proteobacteria and *Haemophilus*.

In an analysis of the sputum microbiota from patients with COPD across four sites in the UK, Wang and colleagues [66] found that the microbiome was heterogeneous in those patients with neutrophilic disease, with two primary communities differentiated by the prevalence of *Haemophilus*: a *Haemophilus*-predominant subgroup and a balanced microbiome subgroup. Patients with a balanced microbiome could temporally switch to both neutrophilic-*Haemophilus*-predominant and eosinophilic states during exacerbations. Temporal changes in the proportions of *Campylobacter* and *Granulicatella* were indicative of switches from neutrophilic to eosinophilic inflammation, keeping track with sputum eosinophilia over time. There were distinct host–microbiome interaction patterns between neutrophilic-*Haemophilus*-predominant, neutrophilic-balanced-microbiome and eosinophilic subgroups. This study suggests that the microbiome could be used to stratify a neutrophilic COPD endotype into subgroups that may benefit from different therapies.

Opron and colleagues [67] examined bronchoalveolar lavage from subjects in the SPIROMICS cohort (never-smokers, smokers without COPD, patients with mild-to-moderate COPD and patients with severe COPD). Microbiome composition alterations (chiefly in the proportions of *Streptococcus*, *Prevotella*, *Veillonella*, *Staphylococcus* and *Pseudomonas*) were found to associate with several clinical features (bronchodilator responsiveness, peak expiratory flow, forced expiratory flow rate 25–75% forced vital capacity), degree of symptom burden and extent of functional impairment. This study highlights that the relationship between the microbiome and airway dysfunction is not only limited to severe disease.

### 2.3. The Microbiome at Exacerbation

In addition to stable disease, exacerbations in COPD are associated with changes in the lung microbiome. Wang and colleagues [68] examined microbiome composition in patients with predominantly GOLD stage II and III COPD and 1–2 previous exacerbations. They found that, prior to exacerbations, samples could be clustered by the dominance of Proteobacteria, Firmicutes or Bacteroidetes phyla. During exacerbations, microbiome diversity decreased and proportions of Proteobacteria and particularly *Moraxella* increased, and these changes correlated with increased blood neutrophil count. When the exacerbation endotype was examined, bacterial exacerbation (defined as a positive bacterial pathogen on routine culture (*H. influenzae*, *M. catarrhalis*, *S. pneumoniae*, *S. aureus* or *P. aeruginosa*) or a total aerobic colony-forming unit count ≥10^7^ cells) was associated with a significant decrease in the proportion of Firmicutes and an increase in the proportion of Proteobacteria compared with eosinophilic exacerbation (defined as sputum eosinophils >3%). At the genus level, there was a decrease in the proportion of *Streptococcus* and an increase in the proportion of *Haemophilus*. There was also a notable decrease in the Proteobacteria:Firmicutes ratio during eosinophilic exacerbations versus all other exacerbation endotypes.

These results were later supported by a larger study by Wang and colleagues [69], in which temporal changes in the microbiome of patients with COPD from the COPDMAP study across three clinical centres were analysed. Across all centres, microbiome composition was similar between stable and exacerbation state, with the exception of a decrease in *Veillonella* at exacerbation. Bacterial exacerbations produced a distinct microbiome profile compared with exacerbations associated with eosinophilic airway inflammation. Changes in the composition of the microbiome during exacerbation were associated with increased exacerbation severity, particularly in eosinophilic patients. Mayhew et al. [70] also found that the stability of the lung microbiome was more likely to decrease over time during exacerbations and in individuals with higher exacerbation frequencies. It was also found that bacterial and eosinophilic exacerbations were more likely to be repeated, whereas viral exacerbations were not. *Haemophilus* and *Moraxella* genera were associated with disease severity, exacerbations and bronchiectasis [71]. In a study from the UK, no temporal differences were found when stable and exacerbation states were compared, consistent with the results of the COPDMAP study. However, this study demonstrated that exacerbations can be sub-grouped into bacterial, eosinophilic and viral/other endotypes, each with significant differences in microbiome composition and clinical characteristics.

Changes in the microbiome following exacerbation have also been found to be associated with mortality. Leitao Filho et al. [72] found that changes were associated with the pathogenesis of acute exacerbations leading to hospitalisation and 1-year mortality. Specifically, increased mortality was associated with lower microbiome diversity and an increased relative abundance of *Staphylococcus* in acute exacerbations. The above studies highlight that the microbiome not only plays a pivotal role in disease progression but could also be used to help understand exacerbation endotypes. There is growing evidence to suggest that COPD exacerbations are heterogenous events that require a personalised approach to treatment.

Taken together, it is clearly demonstrated that the bacterial microbiome in COPD is both individual and highly variable, and that its composition is closely related to disease status and progression, even in mild COPD. It is clear that there are distinct subtypes of COPD exacerbation, which are dependent on both inflammation and microbiome characteristics. The identification of endotypes of COPD exacerbation may be used to guide treatment in the future.

## 3. Interactions between Inflammation, Clinical Characteristics and the Microbiome

### 3.1. Neutrophilic and Eosinophilic Inflammation

Airway inflammation is one of the defining features of COPD, but the inflammatory endotype is heterogeneous across the disease course and between patients; therefore, understanding the nature of inflammation may increase our understanding of COPD and how to treat it. Between 60% and 80% of patients with COPD have predominantly neutrophilic inflammation, which has been associated with increasing exacerbation frequency, lower forced expiratory volume in 1 s, and increased mortality [15,16,17]. Conversely, eosinophil-dominated inflammation accounts for only about 20% of patients and is associated with a higher likelihood of co-existing asthma [18,35]. In addition, exacerbation endotypes are associated with distinct inflammatory profiles [73]. It is therefore pertinent to examine the interplay between the microbiome and the inflammatory status of the airways in COPD. Inflammatory cells, and particularly neutrophils and macrophages, are essential for control of airway infection, whereas airway infection is the primary driver for inflammatory cell recruitment to the airway. Consequently, there is a reciprocal “chicken and egg” interaction between the microbiome and airway inflammation.

Several studies demonstrate this close relationship. Ghebre and colleagues [74] found that the exacerbations of patients with moderate-to-severe COPD could be separated into three biological clusters. All clusters exhibited increased expression of pro-inflammatory mediators. Cluster 1 exhibited increased blood and sputum neutrophils and an increased proportion of Proteobacteria. Cluster 2 exhibited increased blood and sputum eosinophils and an increased proportion of Bacteroidetes. Cluster 3 exhibited an increased proportion of Firmicutes and Actinobacteria. These data suggest an association between sputum pro-inflammatory mediators, cellular and microbiome profiles. Dicker et al. [35] went on to use an integrated clinical, microbiome and proteomic approach to investigate how microbiome profiles influence the pathophysiology of COPD in three predefined sputum microbiome profiles: balanced, Proteobacteria-dominated, and Firmicutes-dominated. In the Firmicutes-dominated group, upregulated proteins were suggestive of a negative regulation of peptidase activity pathway (e.g., cystatin B, cysteine-S, α_1_-antitrypsin). In contrast, in Proteobacteria-dominated patients, there was increased relative abundance of myeloperoxidase, catalase, MMP-9, MMP-8 and neutrophil elastase, all of which reflect neutrophilic inflammation. Pathway analysis indicated that significantly upregulated proteins in the Proteobacteria group were associated with a neutrophil activation pathway. Proteobacteria dominance was associated with increased mortality compared with Firmicutes-dominated or balanced profiles, linking the microbiome with clinical phenotypes and long-term outcomes.

Neutrophilic and eosinophilic inflammation represent broad “umbrella” terms, which hide considerable complexity in cell behaviour. Some endotypes of inflammation deserve special mention. For example, neutrophil extracellular trap (NET) formation is a well-described alternative method of neutrophil antimicrobial defence [75], in which neutrophils extrude webs of decondensed chromatin-containing histones, neutrophil elastase and other granule products that ensnare bacteria. NETs have been observed in the sputum of patients with stable and exacerbating COPD [76,77,78], and COPD disease severity is associated with increased NET-associated neutrophil elastase [76]. Dicker et al. [79] demonstrated that NETs are more abundant in patients with severe COPD, and are associated with more frequent exacerbations, and importantly, reduced microbiome diversity and an increase in *Haemophilus* species. NETs are particularly important as a mechanism in COPD since neutrophil-derived DNA in NETs contributes to the thick sputum production in chronic bronchitis, whereas NET-associated proteases such as neutrophil elastase promote goblet cell hyperplasia and secretion of mucins.

The relationship between bacterial infection and neutrophil levels in COPD is reasonably well described. The relationship with eosinophil levels, however, is unclear, although it is known that eosinophils contribute to a proportion of COPD exacerbations [80]. Kolsum and colleagues [81] showed that low sputum eosinophil levels may be associated with infection with potentially pathogenic bacteria: in patients positive for *H. influenzae*, *M. catarrhalis* and *S. pneumoniae* (defined as ≥10^4^/mL copies of bacterial DNA qPCR target sequence), bacterial load and eosinophil count were inversely related; however, this relationship was only found with sputum eosinophils and not in blood. Conversely, during exacerbations, this inverse relation was found with blood eosinophils in infection-positive patients but not in infection-negative patients. A recent study by Zhou and colleagues [82] also demonstrated significantly increased peripheral blood eosinophil count in patients with COPD acute exacerbations (AECOPD) with pulmonary infection (chiefly *Klebsiella pneumoniae*, *P. aeruginosa*, *Acinetobacter baumannii* and *S. aureus*) compared with AECOPD patients without pulmonary infection.

Taken together, these studies demonstrate that the microbiome is linked to both neutrophilic and eosinophilic inflammatory profiles.

### 3.2. Pneumonia

Although changes in the composition of the microbiome do not necessarily mean infection with potentially pathogenic bacteria, and it is unclear whether such changes lead to increased risk of pneumonia, the latter is theoretically likely as *Haemophilus* infection is the most common cause of pneumonia in patients with COPD.

Pavord and colleagues [83] carried out a meta-analysis of double-blind RCTs of patients with COPD. In 10 trials (10,861 patients) with baseline eosinophil count data, low (<2%) blood eosinophil levels were associated with a small but significant increased rate of pneumonia events compared with patients with eosinophil levels >2%. Further, a network analysis by Martinez-Garcia et al. [84] of patients with moderate-to-severe COPD found that: (i) chronic bronchial infection (CBI) increased the risk of pneumonia in patients with >100 eosinophils/µL, (ii) in patients with <100 eosinophils/µL, risk of pneumonia increased independently of presence or absence of CBI, and (iii) treatment with ICS increased pneumonia risk further in patients with <100 eosinophils/µL and CBI. These two studies suggest that chronic infection, low levels of eosinophils, and low levels of eosinophils together with ICS treatment predispose COPD patients to pneumonia.

Several RCTs, systematic reviews and observational studies have also demonstrated an increased incidence of pneumonia with ICS use in COPD [14]. Consistent with the meta-analysis by Pavord [83], a post hoc analysis of the TORCH study [85]—a double-blind RCT comparing salmeterol, fluticasone propionate (FP), and salmeterol/FP in patients with moderate-to-severe COPD—found a greater rate of pneumonia was observed in the FP and salmeterol/FP arms compared with the salmeterol monotherapy arm. In line with these studies, in two replicate double-blind RCTs of patients with moderate-to-very-severe COPD, radiographically confirmed pneumonia risk was increased with inhaled fluticasone furoate/vilanterol dual therapy versus vilanterol alone [86]. Further, the ETHOS study [87] (a Phase III, randomised trial of budesonide/glycopyrrolate/formoterol versus either glycopyrrolate/formoterol or budesonide/formoterol in patients with moderate-to-very-severe COPD and at least one exacerbation in the past year) found an increase in rates of pneumonia of up to two-fold in the ICS-containing groups versus the LAMA/LABA group. The IMPACT study [88]—a Phase III RCT of fluticasone furoate/umeclidium/vilanterol versus umeclidium/vilanterol—likewise found that the risk of pneumonia was significantly higher (approximately 50% higher) in the ICS-containing group than in the LAMA/LABA group. These studies demonstrate the consistent increased risk of pneumonia infection which can occur with ICS treatment, again emphasising the need for a more personalised approach to treatment.

Evidence for the role of ICS use in increased infection events, pneumonia and exacerbations has been presented. Is it therefore reasonable to hypothesise that ICS use in patients with COPD leads to changes in microbiome composition, in turn leading to the observed adverse events?

## 4. ICS—A Help or a Hinderance? What Effect Does ICS Use Have on the Microbiome?

Recent studies (Table 1) have attempted to analyse the interactions between ICS use (chiefly FP), inflammatory endotype, airway infection, microbiome disordering and pneumonia risk in COPD, bringing together a complex story (Figure 2). The risk of pneumonia is considered a class effect of ICS, and pneumonia has been observed as an adverse effect in randomised studies of all ICS molecules [89]. Nevertheless, different ICS molecules have different pharmacologies and are administered in different doses. Fluticasone is a lipophilic drug and therefore is retained within the airway lumen for longer periods than other drugs such as budesonide. Budesonide is highly soluble in water and is therefore readily absorbed into airway surface liquid followed by translocation into lung tissue. Inside lung tissue, it is conjugated with fatty acids to retain the drug within the lung, therefore reducing the amount of drug in the airway. Kamal and colleagues [90] showed that when airway epithelial cells were infected in vitro with *S. pneumoniae* after treatment at equimolar concentrations of FP, budesonide or beclomethasone dipropionate (BDP), FP and budesonide had marked suppressive effects on pro-inflammatory cytokine release. Similar effects were observed in mice infected with *S. pneumoniae*, with greater suppression of host immunity with FP and budesonide compared with BDP. In addition, bacterial loads in the lungs of infected mice were increased with FP and budesonide but not with BDP. Heijink and coworkers [91] compared the effects of budesonide and FP on human bronchial epithelial cells in response to a viral mimic or cigarette smoke extract and found similar effects on preventing pro-inflammatory cytokine secretion. However, at the level of the epithelial barrier function, which is reduced by infection or cigarette smoke, budesonide protected the epithelium to a greater degree than FP. Provost and colleagues [92] showed that both ICS reduced pro-inflammatory cytokine release from monocyte-derived macrophages equally, but similarly found that budesonide was more effective than FP in preventing bacteria-induced receptor suppression in macrophages. Although pneumonia is a class effect of ICS as stated above, these studies demonstrate the different effects of ICS within the class on markers of inflammation and host immunity.

As well as the above effects on the host, it should be considered whether corticosteroids can directly impact bacteria; however, studies are lacking. As noted, *H. influenzae* is the most frequent pathogen isolated in severe COPD, and studies suggest that through indirect effects on the host, FP and budesonide inhibit intracellular persistence of *H. influenzae* [98,99]. Budesonide has also been shown to promote invasion of epithelial cells by *P. aeruginosa* [100]. These data suggest that ICS may influence changes in the microbiome by promoting the persistence of some bacteria and opposing the persistence of others. Much less is known of the direct effects of corticosteroids on bacteria under physiological conditions.

Due to the relatively new nature of the field, studies specifically powered to assess the impact of ICS on the lung microbiome in COPD are few. Clues may be gleaned from studies in other respiratory diseases. For example, in asthma, it has been demonstrated that inflammation is related to the lung microbiome composition, and changes in the composition were associated with FP use [36]. A pilot study in rhinosinusitis also demonstrated that increases in the diversity of the nasal cavity microbiome resulted from the use of the intranasal steroid mometasone furoate [37].

Contoli and colleagues [93] were the first to carry out a study powered to investigate the effect of long-term treatment with an ICS added to a LABA on sputum bacterial load in patients with stable COPD. This key, proof-of-concept, prospective trial randomised patients with stable moderate COPD to receive either salmeterol/FP combination therapy or salmeterol alone for 12 months. Compared with salmeterol alone, 12 months of treatment with the combination therapy resulted in a significant increase in sputum bacterial load, modification of the sputum microbial composition and increased airway load of potentially pathogenic bacteria. Importantly, increased bacterial load and increased proportion of *S. pneumoniae* and *H. influenzae* were only observed in patients with lower (≤2%) sputum or blood eosinophils, and not in patients with higher baseline eosinophils. In each group (salmeterol/FP vs. salmeterol alone), there was no significant difference in microbiome composition at baseline versus 1 year of treatment. However, when comparing salmeterol/FP versus salmeterol alone, there was a significant increase in diversity after 1 year of treatment, and an increased proportion of Firmicutes and *Candida* species, with a significant reduction in Proteobacteria. At the species level, a relative increase in the proportion of *S. pneumoniae* and *H. influenzae* was observed in the dual therapy group after 1 year.

Recently, Leitao Filho and colleagues [94] compared the effects of treatment with budesonide/formoterol and FP/salmeterol against formoterol only on the airway microbiome of patients with stable COPD. After 12 weeks’ treatment with FP/salmeterol, a significant reduction in airway microbiome diversity was observed compared with formoterol alone, and a greater number of changes in microbiome from baseline compared with both formoterol alone and budesonide/formoterol groups. The greatest changes in bacterial relative abundance in the FP/salmeterol group included a decrease in the abundance of *Haemophilus*. A significant decrease in the Proteobacteria:Firmicutes ratio was observed, which was driven by a non-significant increase in abundance of Firmicutes and the above significant decrease in the abundance of *Haemophilus*. Overall bacterial load was not significantly different between treatment groups. These results present some differences from those of Contoli and colleagues, who, in the FP/salmeterol group, showed an increased bacterial load, an increased proportion of *H. influenzae* and increased diversity. These differences may be due to sampling, FP dose and the LABA comparator selected: Contoli sampled sputum, whereas Leitao Filho employed bronchoscopy; the dose of FP used in Contoli’s study was lower; and Contoli selected salmeterol as the LABA comparator whereas Leitao Filho selected formoterol. Both studies, however, did show a decrease in the Proteobacteria:Firmicutes ratio.

Other studies have broadly agreed with Contoli’s study, although they were not specifically powered to investigate the impact of ICS on the bacterial microbiome. A study by Singanayagam [95] of patients with stable COPD also found that patients taking FP had a significantly higher abundance of *Streptococcus*, with increased overall bacterial load and diversity compared with patients not taking ICS. A limitation to this analysis is its cross-sectional nature which cannot infer causality. Therefore, to examine the mechanisms of this apparent ICS-induced proliferation of *Streptococcus*, cellular and mouse models of lung infection with *S. pneumoniae* were treated with FP. An increase in the abundance of *S. pneumoniae* and impaired bacterial clearance was observed, which was associated with downregulation of the antimicrobial peptide cathelicidin. In the same study, Singanayagam went on to examine the mechanism of FP-mediated downregulation of cathelicidin and it was found that augmentation of the protease cathepsin D led to ICS-induced degradation of cathelicidin. It should be caveated, however, that mouse models may be poorly predictive of effects in humans. The majority of Streptococci in the COPD microbiome are oropharyngeal rather than *S. pneumoniae* and so the animal models may not be directly linked to the human observations. In an investigation of bacterial load and microbiome composition in stable disease and exacerbations, Garcha and colleagues [96] demonstrated a dose–response relationship between ICS dose, bacterial load and severe airflow limitation at a stable state. However, again, bias may exist in cross-sectional studies such as these, as patients on ICS usually have more severe disease. Huang and colleagues [97] found that chronic oral steroid use in COPD altered the microbiome composition, with a trend towards increased diversity, and in patients treated with oral steroids plus antibiotics, there was a significant increase in Proteobacteria. The authors suggested that alterations in the airway microbiome may be due to steroid-induced immunosuppression. In a similar study, Wang et al. [68] examined COPD patients treated with oral corticosteroids, antibiotics and a combination of both at stable state, exacerbation, 2 weeks post-therapy and 6 weeks’ recovery. In patients taking oral corticosteroids, a decrease in diversity was found, and increased Proteobacteria over Firmicutes (at the genus level, an increase in *Haemophilus* and *Moraxella*, and a decrease in *Streptococcus*). The apparent contradiction in which inhaled steroids promoted increased *Streptococcus* in the study by Singanayagam but oral corticosteroids caused an increase in Proteobacteria and decreased *Streptococcus* in the studies by Wang and Huang, has not been explained. In the study by Dicker and colleagues [79] described earlier in this review, in which a positive correlation between NET formation and fluticasone/salmeterol dual therapy was demonstrated, an in vitro analysis demonstrated that therapeutically relevant doses of fluticasone inhibited neutrophil phagocytosis of fluorescently labelled *E. coli*. As it has previously been shown that inhibition of neutrophil phagocytosis leads to NET formation [101], and that NETs are a less effective means of bacterial killing [102], the authors speculate that this may be a mechanism by which ICS use in COPD may inhibit phagocytosis, leading to the observed decrease in bacterial diversity and increased abundance of *Haemophilus*. Scott and colleagues [103] also showed that phagocytosis was impaired by fluticasone, beclomethasone and budesonide in a dose-dependent manner, with fluticasone having the greatest inhibitory effect.

Taken together, these studies suggest that ICS use may promote increases in bacterial load through impaired neutrophil phagocytosis, impaired neutrophil apoptosis and increased NET formation.

## 5. Conclusions

ICS is indicated in patients with frequent exacerbations and an eosinophilic endotype. The evidence presented in this review suggests that ICS use in patients with a neutrophilic endotype may lead to changes in the composition of the lung microbiome that could potentially explain the increased risk of pneumonia and other adverse events seen in neutrophilic patients. This review therefore bolsters the GOLD recommendations to appropriately target ICS based on inflammatory endotype, and to avoid inappropriate prescription of ICS in patients without a history of exacerbations or with neutrophil dominant disease. Point-of-care or near-patient tools that can detect neutrophilic inflammation, the use of blood eosinophils or detection of pathogens such as *H. influenzae* and *Streptococcus* may aid precision medicine approaches in the future in identifying which patients are most likely to achieve optimal risk:benefit in COPD.

## Figures and Tables

**Figure 1 biomedicines-09-01312-f001:**
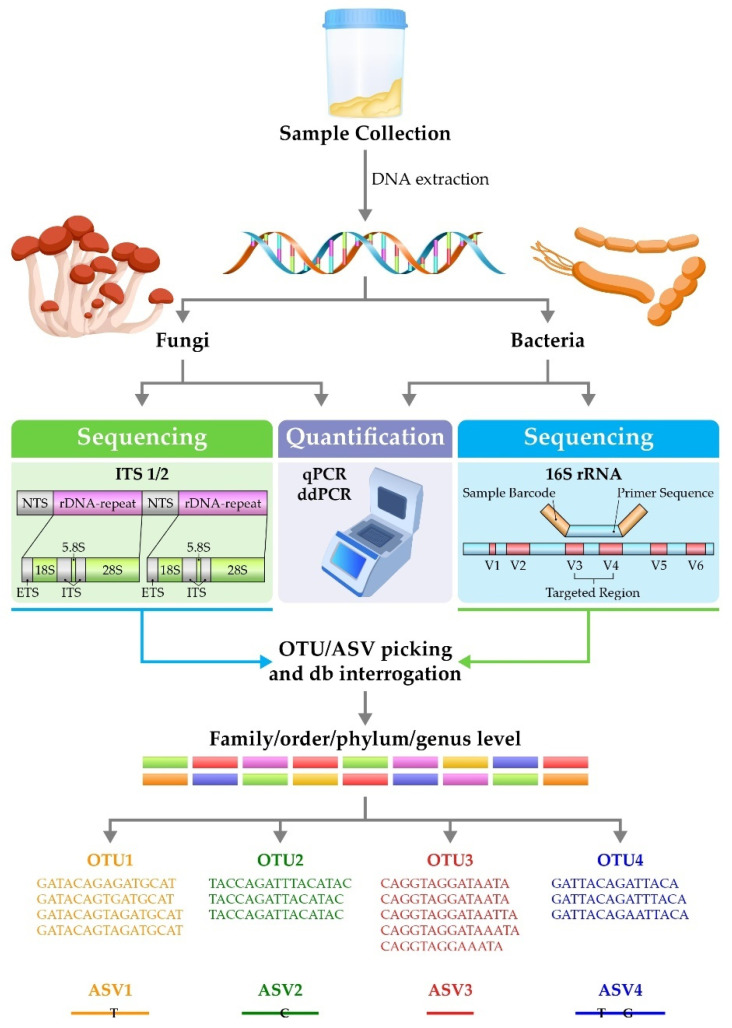
Analysis of the lung microbiome. Abbreviations: ASV: amplicon sequence variant; db: database; ddPCR: droplet digital quantitative polymerase chain reaction; ETS: external transcribed spacer; ITS: internal transcribed spacer; NTS: non-transcribed spacer; OTU: operational taxonomic unit; qPCR: quantitative polymerase chain reaction; V: variable region.

**Figure 2 biomedicines-09-01312-f002:**
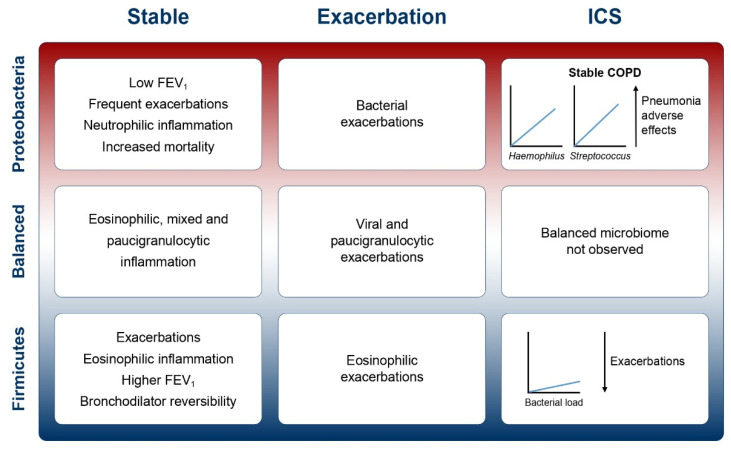
Overall associations between the spectrum of microbiome composition, inflammatory endotype, ICS use and disease phenotype in COPD. Abbreviations: FEV_1_: forced expiratory volume in 1 s; ICS: inhaled corticosteroids. In Proteobacteria-dominant, neutrophilic patients, ICS use may promote increased bacterial load and increased *Haemophilus* and *Streptococcus* [93], while in Firmicutes-dominant, eosinophilic patients, lesser changes in the airway microbiome were observed, potentially linking to their lower risk of infective adverse events [83,93].

**Table 1 biomedicines-09-01312-t001:** Summary of studies examining the effects of ICS on the lung microbiome.

Authors	Year	Study Population	Treatment	Endpoints	Observations
Turturice [36]	2017	Adult atopic asthma, intermittent or mild/moderate persistent symptoms (*n* = 13); age-matched non-asthmatic controls (*n* = 6)	6 weeks’ FP vs. before treatment	Inflammatory markers (cytokine + chemokine panel) and metagenomic sequencing of lower airway microbiome in BAL	Reductions in *S. pneumoniae*, *N. meningitis*, *E. faecium*, *E. faecalis*. Associated with decreased MIP-1β, increased IL-2
Ramakrishnan [37]	2018	Adult chronic non-infectious rhinitis males (*n* = 4); healthy female (*n* = 1)	Non-infectious rhinitis males: mometasone furoate nasal spray QD for 1 month healthy female: BID topical mupirocin decolonisation	Serial nasal cavity swab microbiome analysis over 8 weeks (16S rRNA gene)	Increased abundance of staphylococci; reduced abundance of *Moraxella* and streptococci. Increase in diversity in 2/4 subjects
Contoli [93]	2017	Steroid-naïve, stable moderate COPD (*n* = 60) on treatment with SAL	1:1 SAL/FP:SAL BID 12 months	Sputum bacterial load, microbiome composition (16S rRNA gene)	Increased bacterial load, an increased proportion of *S. pneumoniae*, *H. influenzae* with low blood/sputum eosinophils. Increase in diversity, an increased proportion of Firmicutes, *Candida*; the reduced proportion of Proteobacteria
Leitao Filho [94]	2021	Adults (*n* = 63) with stable moderate-to-severe COPD, 4-week ICS washout, 4-week run-in with FORM	BUD/FORM or FP/SAL vs. FORM	Bronchoscopy bacterial load, microbiome composition, microbiome changes vs. clinical parameters	In FP/Sal group, reduction in diversity, greater number of changes in microbiome from BL, decreased abundance of *Haemophilus*, decreased Proteobacteria:Firmicutes ratio
Singanayagam [95]	2019	Stable mild-moderate COPD: current ICS use (*n* = 10); non-use of ICS (muscarinic antagonist; SABA/LABA; *n* = 13)	FP, BUD, BD	Sputum bacterial load, microbiome composition (16S rRNA gene)	Increase in abundance of *Streptococcus*; increased bacterial load and diversity
		Patients reporting exacerbations: current ICS use (*n* = 11); non-use of ICS (*n* = 16)	FP, BUD, BD	Sputum hCAP18, BAL cathepsin D concentration	Suppressed hCAP18 concentrations; increased cathepsin D concentrations
		Mouse model of COPD; WT mice	FP vs. pre-FP	*S. pneumoniae* load, cathelicidin-related AMP concentration, cathepsin D concentration in BAL, whole lung, blood	Increased *S. pneumoniae* load; reduced concentrations of cathelicidin-related AMP; increased cathepsin D concentrations
		Cathelicidin knock-out mouse	FP vs. control	Bacterial load; *S. pneumoniae* load in BAL	No effect of FP on cathelicidin knockouts
		BEAS2B bronchial epithelial cells	FP vs. control	hCAP18 concentration; cathepsin D concentration	Suppressed hCAP18 concentrations; augmented cathepsin D induction
		COPD primary bronchial epithelial cells	FP vs. control	hCAP18 concentration	Suppressed hCAP18 concentrations
Garcha [96]	2012	Stable COPD GOLD stages II–IV (*n* = 134)	*n* = 47 using ICS (median [IQR] beclomethasone-equivalent dosage 2000 (640–2000) mg daily)	Sputum bacterial load; severe airflow limitation	Higher airway bacterial load associated with higher ICS usage and more severe airflow limitation
Huang [97]	2014	COPD patients with bacterial infection (*n* = 12)	Antibiotics only vs. oral corticosteroids only vs. both	Sputum microbiome composition (16S rRNA gene)	Oral corticosteroids alone: increased proportion of Proteobacteria, Bacteroidetes and Firmicutes, particularly *Enterobacteriaceae*, *Lachnospiraceae*, *Burkholderiaceae*, *Neisseriaceae* Oral corticosteroids plus antibiotics: increase in Proteobacteria
Wang [68]	2016	Stable and exacerbative COPD patients (*n* = 87)	Antibiotics only vs. oral corticosteroids only vs. both	Sputum microbiome composition (16S rRNA gene)	Oral corticosteroids alone: decreased diversity; increased proportion of Proteobacteria; decrease in *Streptococcus*, increase in *Haemophilus* and *Moraxella*

Abbreviations: AMP: antimicrobial peptide; BAL: bronchoalveolar lavage; BD: beclomethasone dipropionate; BID: twice daily; BL: baseline; BUD: budesonide; COPD: chronic obstructive pulmonary disease; FLUT: fluticasone; FORM: formoterol; FP: fluticasone propionate; GOLD: Global Initiative for Chronic Obstructive Lung Disease; hCAP18: human cathelicidin; ICS: inhaled corticosteroids; IL-2: interleukin 2; IQR: interquartile range; LABA: long-acting β_2_-agonist; MIP-1β: macrophage inflammatory protein 1β; QD: daily; SAL: salmeterol; SABA: short-acting β_2_-agonist; WT: wild-type.

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
