# Peer review of "Inhaled Corticosteroids and the Lung Microbiome in COPD"

_biomedicines, 2021, doi:10.3390/biomedicines9101312_

Round 1
Reviewer 1 Report
General: Keir et al have written a review summarizing what is known to date on the effect of inhaled corticosteroids on the respiratory microbiome in COPD. This is an important scientific topic with practical clinical implications today given nuanced decision making that must weigh benefits of improved airway inflammation and lessened exacerbations (particularly for those with eosinophilic inflammation), but increased risk of pneumonia. Overall, I think this is a very nicely constructed review and the sections on COPD are quite well done. I do think given it is the stated focus of the review, there can be a bit more done to dissect out the impacts of inhaled corticosteroids on the microbiome while acknowledging that there is less direct evidence available here. My specific comments below.
Major comments:
- Mechanisms of ICS and COPD / microbiome: I think in general, given there is less data directly guiding the discussion on how the microbiome is impacted by inhaled corticosteroids, it is worth giving a more robust discussion of what is known about ICS and how they affect mucosal immunity and bacteria. This may help frame a putative understanding of the impact on the airway microbiome and COPD. (A) I think section 4 might need a bit more background on the different inhaled corticosteroids such as differences within the class and differential dose effects. (B) Relatedly, the introduction has a nice paragraph on different impact of ICS on PMN and eosinophils, but this might be expanded upon. Effects on epithelial cells? Alveolar macrophages? (C) While less well described how might corticosteroids directly impact bacteria? How do bacteria impact steroid pharmacodynamics? (D)There is some evidence suggesting the type of inhaled corticosteroid might differentially impact pneumonia risk (PMID 34267661) with the lipophilic fluticasone seems to confer highest risk. Overall, I feel there is an opportunity to improve section 4 to delineate a framework for understanding how ICS impacts the microbiome in COPD.
- Relationship between ICS, COPD, and microbiome: As the authors note, it is hard to assess causal relationships in a complex disease such as COPD, let alone how ICS might alter these dynamics. That said I think Figure 2 might be able to be improved to clarify some of the broader themes. For instance, do the rows apply across all columns? It is unclear if Proteobacteria associations apply to all three columns top row. There is no information in balanced in 2 of three columns. How does ICS relate to stable and exacerbation – as depicted here it is independent. I understand the main intent but think it is a bit too information dense (perhaps splitting into different figures/panels?) and clarity could be improved in understanding relationships across columns and rows.
- Figure 1 comments: (A) Consider including other modalities to assess bacterial biomass such ddPCR (Combs, Lancet Resp Med 2021). (B) There are other emerging ways to cluster bacteria such as amplicon sequence variants (ASV). Clearly some debate about best methods but consider including. (C) Not always Genus level – frequently lower resolution to family/order/phylum and rarely better to species level. (D) Primer sequence is a bit misleading as it seems to be indicating the whole amplicon between the barcodes.
Minor comments:
-Table 1: consider consistently stating the sample types used in the studies
-Line 115: not always genus level – frequently less, rarely more granular.
-Line 116: would say within the sample, not necessarily the population.
-Lines 119-121 asthma severity interestingly not always associated with low diversity. (cites)
-Line 141: agree with the point but would argue it might be more accurate to say compositional (relative abundance) here.
-Line 163: meta-analysis
-Suggest a line explaining how bacterial and eosinophilic exacerbations were defined.
-In general multiple long paragraphs made for dense reading. Suggest breaking up blocks of text where feasible.
Reviewer 2 Report
The review by HR Keir, M Contoli, and JD Chalmers reports the state of the art regarding the microbiome in COPD and discusses the optimal use of ICS to approach personalized pharmacological management. The manuscript is well structured and pleasant to read. The subject is original and of great interest to the readers of biomedicines and the actors of respiratory researches. The references are up to date and sufficient. Two figures and one table complete and complement the text. I have only minor comments that the authors should consider:
- Figure 1: is there a reason why there are 4 OTU mentioned?
- Lines 132-133: it is “being increasingly used” according to the authors but only one reference is mentioned (n°43). Is it possible to expand?
- Concerning 2.1:
- is there a proposed cartography of the lung microbiome (in terms of quantity and relative abundance)? What about a differential microbiome according to the lung sub-location (i.e. bronchi vs bronchioles vs parenchyma, etc.)?
- It would be useful to precise (when possible) the exact status of the “healthy lung” since bacterial load and proportions may directly be impacted by the clinical status of the patient at the moment of the biological specimen collection.
- Concerning 2.2: what about the association of the microbiome features with the severity of COPD?
- The number of patients in the study population is not always mentioned in the text. Is there a reason? The description of the main observations could be more consistent throughout the manuscript, particularly before part 4.
- Figure 2 is mentioned in the text (line 362) but is not described in the manuscript. In addition, it could be improved to gain clarity, while increasing the number of details to discuss the associations. I find it a little too simplistic.
Reviewer 3 Report
Thank you for submitting an excellent review. There is nothing to comment on in the draft.
Round 2
Reviewer 1 Report
I think the authors have done an excellent job responding to comments and feel the review is comprehensive and essential reading. I do not have further comments or suggestions as I feel the authors have made significant improvements where they were able and otherwise made sound justifications for their decisions.